# Seminal Plasma Glycoproteins as Potential Ligands of Lectins Engaged in Immunity Regulation

**DOI:** 10.3390/ijerph191710489

**Published:** 2022-08-23

**Authors:** Beata Olejnik, Mirosława Ferens-Sieczkowska

**Affiliations:** Department of Biochemistry and Immunochemistry, Wroclaw Medical University, 50-368 Wroclaw, Poland

**Keywords:** glycosylation, lectins, infertility, immune response, regulatory T-cells

## Abstract

Environmental pollution, chronic stress, and unhealthy lifestyle are factors that negatively affect reproductive potential. Currently, 15–20% of couples in industrialized countries face the problem of infertility. This growing health and social problem prompts researchers to explore the regulatory mechanisms that may be important for successful fertilization. In recent years, more attention has been paid to male infertility factors, including the impact of seminal plasma components on regulation of the female immune response to allogenic sperm, embryo and fetal antigens. Directing this response to the tolerogenic pathway is crucial to achieve a healthy pregnancy. According to the fetoembryonic defense hypothesis, the regulatory mechanism may be associated with the interaction of lectins and immunomodulatory glycoepitopes. Such interactions may involve lectins of dendritic cells and macrophages, recruited to the cervical region immediately after intercourse. Carbohydrate binding receptors include C type lectins, such as DC-SIGN and MGL, as well as galectins and siglecs among others. In this article we discuss the expression of the possible lectin ligands, highly fucosylated and high mannose structures, which may be recognized by DC-SIGN, glycans of varying degrees of sialylation, which may differ in their interaction with siglecs, as well as T and Tn antigens in O-glycans.

## 1. Introduction

Problems with procreation are now becoming a serious medical and social issue. Currently, it is estimated that 15–20% of couples in industrialized countries may struggle to conceive. A huge role in this rising problem is attributed to the environmental factors, associated with ubiquitous toxic pollutants, affecting gamete production and quality [1,2,3,4]. Such factors impact also the process of fertilization, embryo development, and normal pregnancy. The infertility problem always concerns both partners who should undergo diagnostics and therapy. Nevertheless, some specific factors can be attributed to either the female or the male partner. Evaluation of the male infertility factor is still often limited to the standard examination of the semen, focused mainly on the number, morphology, and motility of sperm. Such results can hardly be considered a reliable assessment of a male’s reproductive potential [5,6,7,8]. Currently, more attention is paid to the analysis of the events accompanying fertilization at the molecular level, which should allow understanding these extremely complex processes. Among some newly considered factors, the role of extracellular seminal components, i.e., seminal plasma, in the regulation of the fertilization process is worth examining [9,10,11,12]. Seminal plasma components, including TGF-β and E-series prostaglandins are involved in triggering post-coital acute leucocyte reaction; thus, recruitment of immune cells to the cervix area as we have briefly reviewed in our former article [13]. Further interactions lead to proliferation of regulatory T-cells, participating in the achievement of tolerance for male-originated antigens of sperm and embryo. According to the fetoembryonic defense hypothesis [14] the other mechanism to suppress maternal immunity may involve lectins present on the surface of these immune cells recruited to the cervix area and their seminal counterparts. In this review, we want to address the importance of glycosylation, a common post-translational modification of secreted proteins, as well as its possible impact on interactions with lectins within the female genital tract, and their possible contribution to the regulation of the fertilization process.

## 2. Glycosylation at a Glance

The post-translational attachment of oligosaccharides to the polypeptide backbone and therefore decoration of protein surface with glycans concerns more than half of all proteins. When we consider secretory proteins, this percentage is much higher, because glycosylation works as a “traffic signature”, facilitating the foundation of the protein on the cell surface or its extracellular secretion [15,16,17]. Oligosaccharides can be attached to the protein core by the two types of linkage: N- or O-glycosidic. In the first case, the acceptor is amide group of asparagine, and the first sugar residue is always N-acetylglucosamine. The oligosaccharide in this pathway is synthesized on a lipid carrier (dolichyl phosphate) anchored in the endoplasmic reticulum membrane, and then transferred in the form of saccharide unit (Glc_3_Man_9_GlcNAc_2_-N) to the specific N-glycosylation acceptor sequence (Asn-X-Ser/Thr) of the newly synthesized protein. Such a glycan later undergoes extensive reconstruction in the cisterns of Golgi. Mature glycans of complex-type (typical for vertebrates) do not contain glucose and mannose in the external antennae. These two monosaccharides are replaced with the LacNAc sequence, capped with sialic acid (Neu5Ac) and optionally fucose (Figure 1A).

In mucin-type O-glycans, the sugar unit (GalNAc) is accepted directly by the hydroxyl of serine or threonine of the protein backbone [16,18]. Next, a set of glycosyltransferases attach subsequent monosaccharide residues and build the structures known as cores 1–4, and later their elongated derivatives (Figure 2A). The final product of the glycosylation pathway—the structure of mature glycan—depends on many factors, including the availability of the protein to be glycosylated, monosaccharide substrates activated on the nucleotide carriers as well as the activity of numerous, sequentially acting Golgi glycosyltransferases. The result of such a complex process is a huge variety of glycoforms of the resulting protein-variants differing in detailed structure of oligosaccharide chains. It is extremely important that the basic glycosylation pattern remains mostly constant (with all the diversity of glycoforms) until we consider a healthy individual, and the remarkable modifications are often associated with the pathogenesis of some diseases, e.g., in cancer or inflammation [19,20,21,22]. A “healthy” glycosylation pattern may also be tissue-specific and related to the developmental stage of the individual. For more detailed information on the glycosylation pathways the reader is referred to [23].

As a reference point, the basic pattern of glycosylation of serum glycoproteins of a healthy human is most often used. For N-linked glycosylation, this implies a predominance of complex-type biantennary glycans terminated with a sialic acid residue (Figure 1A) [24,25]. Highly branched glycans (tri-and especially tetra-antennary) and glycans containing fucose are less abundant. In the serum of a healthy person, the forms deprived terminal sialic acid are absent in practice, as this feature is a signal for glycoprotein removal in the liver [26,27,28]. Additionally, no significant amounts of high-mannose and hybrid glycans are observed in vertebrates. These forms, evolutionarily older, dominate in fungi, plants, and invertebrates [29].

## 3. Glycosylation and Diseases

Already in the 1990s, it was noticed that significant changes in the glycosylation pattern are associated with disease processes and in many cases may even be considered disease markers [30,31,32]. Until now, certain features of the oligosaccharide structures were indicated as disease related (Figure 1B and Figure 2B), including: The degree of sialylation and the method of Neu5Ac residue attachment [33,34].The degree of fucosylation and the type of bond used to attach this monosaccharide [20,35].Branching of N-glycans, including the presence of “bisecting” GlcNAc and the fourth, β1,6 linked antenna [36,37,38].Leaving O-glycans in a truncated form [39,40,41].

Altered glycosylation has been often observed and documented in relation to inflammation as well as neoplastic transformation and metastasis [42,43]. The altered glycoform patterns were observed both in tumor tissues and in the form of so-called circulating markers, presented by blood plasma glycoproteins. Numerous studies in this area have enabled identification of glycoepitopes collectively referred to as TACA—tumor associated carbohydrate antigens [44,45], including the sLe^X^, Le^Y^, and related antigens and the above-mentioned glycans with the β1,6 antenna and bisecting GlcNAc, as well as the truncated Tn and T antigens of mucin-type O-glycans (Figure 2B).

## 4. Lectin–Protein Receptors for Sugar Structures and Lectin–Carbohydrate Axes in Immunity Regulation

The oligosaccharide sequence can be specifically recognized by receptor proteins called lectins or carbohydrate-binding proteins (CBPs) [46,47,48,49,50]. The research on their properties including sugar binding specificity, as well as the role of lectin-carbohydrate interactions in physiological and pathological processes, is nowadays a rapidly exploding area [47,51,52,53,54]. Proteins of this type can be divided into many categories [55], whereas in this article we focus on three of them: C-type lectins, galectins, and siglecs. 

The C-type lectins group is distinguished by their common demand of calcium ions for ligand binding [56,57,58,59]. This group mostly includes proteins that reside on the outer surface of cells. Galectins are soluble proteins, specific for β-galactosides [60,61,62,63,64]. The siglecs, belonging to the immunoglobulin superfamily (I-type lectins), recognize terminal sialic acid residues [65,66,67,68]. Immune cells are especially rich in lectin residents. Many types of cells create the immune system, with different origins, degrees of maturity and differentiation, and functions. Their orchestrated operation allows for mutual communication, transfer of information on the status of the encountered antigens, and extremely precise control of the immune response. Dendritic cells (DCs) are believed to be the major players in regulating this response [69,70,71]. Depending on their subtype (origin), as well as the stimulating factors present in the microenvironment, dendritic cells can induce a pro-inflammatory response or trigger tolerogenic reactions. Peripheral dendritic cells capture antigens, internalize them and, after processing, present them on the surface in the context of MHC II to T-cells in the lymph nodes. Their response aims to recognize the pattern of “foreign” antigens presented by pathogens (PAMPs, pathogen associated molecular pattern) and self, but somehow damaged cells (DAMPs, damage/danger associated molecular pattern) in order to be destroyed and removed by the effector cells. On the other hand, a similar mechanism prevents effector response to self-associated antigens (SAMPs) [72,73]. In some situations, the immune system may not achieve the appropriate balance. The result may be an undesired inflammatory reaction towards the own cells or tissues, as in autoimmune diseases and allergies, but also the lack of demanded reaction, which results in the evasion of pathogens or tumor cells from the control of the immune system and, consequently, disease progress. In the latter case, we most often deal with overexpression of SAMPs (self-associated) antigens on the cell surface. To accomplish these tasks, the surface of the DCs is decorated with numerous PRRs (pattern recognition receptors), which include TLRs, NLRs, RLRs and—as mentioned earlier—numerous lectins. When DCs remain immature, their further interactions support tolerogenic reaction. Mature DCs, fully differentiated, induce an effector T-cell response [74,75,76]. The above-mentioned general directions work through complex signaling pathways, supported with many other factors. For more detailed information on DCs roles in immunity the reader is referred to the reviews of van Vliet et al. [77], Tisch [78], Waisman et al. [79], and many others.

To discuss the properties and functions of carbohydrate binding protein receptors, let us start from C-type lectins. This extremely diversified group of lectins—in opposite to galectins and siglecs—comprises different, non-homological proteins, mostly but not always calcium-dependent, including DC-SIGN, MGL, MBP, collectins, selectins, dectins, and many others [59,75,80,81]. Only the two of them—DC-SIGN and MGL—were selected for the current review.

### 4.1. DC-SIGN

Dendritic Cell-Specific Intercellular adhesion molecule-3-Grabbing Non-integrin (DC-SIGN), also known as CD-209, is type-II membrane receptor presented by macrophages and dendritic cells (DCs). DC-SIGN is able to bind the two different kinds of glycans: high mannose type and highly fucosylated. The interaction is involved in recognition of pathogens and their intake by endocytosis [82].

Structurally, the protein consists of an extracellular part including carbohydrate recognized domain (CRD) and neck repeated regions, followed by transmembrane fragment and cytoplasmic domain. The latter includes internalization motifs, i.e., tyrosine-based motif (YYY), triad of acidic amino acids (EEE) and di-leucine motif [80,83]. The single CRD of DC-SIGN is monovalent for mannose or fucose; thus, for the binding of branched glycans the lectin receptor creates tetramers in the cell membrane with the coiled-coil α-helices of neck repeated fragments [84,85,86,87]. CRD and the tandem repeats in outer domain are responsible for the further formation of membrane clusters (Figure 2). Stabilization of these structures is due to binding with extracellular matrix (ECM) components, e.g., glycosaminoglycans or proteoglycans, as well as with transmembrane adaptor proteins (TRAPs) on the surface of the other cells [88,89]. While the cytoplasmic domain does not participate in cluster formation, it may be important for its final shaping [89,90]. While the non-occupied cluster of DC-SIGN microdomains is floating in the cell membrane, after the interaction with its counterpart the whole structure is finally stabilized. CRD avidity for various glycans differs and depends on spatial arrangements of large, branched carbohydrates [82,91,92]. Experimental divalent chimeric DC-SIGN-Fc showed high affinity for the glycans with high mannose content, resulting from α anomericity of the central mannose in the glycan structure [91]. The binding strength decreases with lowered number of mannose residues [93,94]. In opposite, attraction for the complex fucosylated oligosaccharides was diminished among Lewis^b^, Lewis^y^, Lewis^a^, and Lewis^x^, respectively [91] The speed of equilibrium achievement and dissociation step was higher while binding the oligomannose glycan, whereas for Lewis^x^ type glycan time of association and dissociation was longer [91]. DC-SIGN was not able to bind sialylated glycans [82].

Immune response initiated by DC-SIGN may result in two different ways depending on glycosylation pattern (Figure 3), but also the size and space arrangement of the antigen. When the mannose-presenting carbohydrate is associated with the DC-SIGN, the immune response is shifted toward the inflammatory pathway, whereas the DC-SIGN interaction with fucosylated glycans, such as Lewis-type, was shown to limit the pro-inflammatory cytokine secretion which leads to attenuation of the immune response [87,95].

After binding of high-mannose glycans, the interaction between dendritic cells and neutrophils results in the T-cell proliferation and immune response for the pathogen entry. DC-SIGN can bind through carcinoembryonic antigen-related cell adhesion molecule 1 (CEA-CAM1) on neutrophil surface [96] or ICAM-3 on T-lymphocytes [69].

Immunological evasion, observed in persistent pathogen contagion, as well as tumor metastasis is supposed to be the interaction with fucosylated glycans. The silencing of immune response mediated by DC-SIGN on dendritic cells was described for some pathogens (e.g., HIV, *Schistosoma*, *Helicobacter pylori*) [55,95,97] or tumor metastasis [98]. HIV infection to human cells is due to the interaction of highly glycosylated proteins with an envelope rich in Le^x^ type glycans (gp41, gp120) with DC-SIGN on dendritic cells and then transport of the virus to lymph nodes rich in CD4^+^ [14]. Although the mechanism that allows DC-SIGN to distinguish between high-mannose and fucosylated glycans is not clear, it was proved that binding of these different glycan categories results in different signal transduction into the cell and activation of either pro- or anti-inflammatory pathways.

### 4.2. MGL

Another C-type lectin expressed on immune cells is CLEC10a, known also as macrophage galactose-type lectin (MGL). The lectin presents unique specificity only to GalNAc [55,81]. The acetyl moiety of galactosamine is important for the recognition process, so the lectin generally does not bind common galactose-presenting glycans and epitopes such as LacNAc (Gal-β1,4-GlcNAc). On the other hand, binding is not hampered with the substitution of sialic acid at the C6 position of galactose and thus both Tn antigen and its sialylated form are the best ligands for the lectin. Recently, a binding of unique LacdiNAc disaccharide has been also reported (Figure 4). All three sugar structures are absent (or extremely rare) under normal conditions, but their presentation increases after neoplastic transformation. The Tn and sTn antigens are typical MUC1 glycoforms associated with tumor cells. Additionally, LacdiNAc epitopes have been reported in tumor glycoproteins, including prostate cancer [99,100]. Contrary to many other lectins, no cluster effect was observed in the interaction of MGL with the Tn antigen.

It is not entirely clear what the role of MGL is in establishing self-tolerance. Presented on the surface of macrophages and dendritic cells, it is highly expressed in tolerogenic subpopulations of these cells [55]. This lectin appears to play a protective role in prolonged, persistent inflammation and autoimmune diseases by preventing excessive tissue damage. MGL interactions participate in redirecting the activity of DCs to reduce the activity of effector T-cells and even induce their apoptosis [55]. The lectin may be also involved in the retention of immature DCs: the blockage of MGL with specific antibodies increased the migration of DC to the lymph nodes, where they induced an effector response. Cancer cells appear to target MGL interaction by displaying an increased number of truncated O-glycans, Tn and sTn, and use this mechanism to increase their own survival.

MGL rarely recognizes pathogens and modulates the response to them, due to the rarity of the recognized epitope in pathogen glycans, with few exceptions including *Mycobacterium tuberculosis* [101].

### 4.3. Galectins

This group of lectins has been distinguished based on their specificity to oligosaccharides containing beta-galactose. Uniquely, these lectins are not bound to cell membranes, instead they can reside and function both intracellularly and in the form of soluble proteins secreted into the extracellular space. For the current considerations, we will focus only on the latter activity of galectins. Structurally, galectins can be classified into three types [61,102]. The prototypical galectins (Gal-1, Gal-2, Gal-5, Gal-7, Gal-10, Gal-11, Gal-13, Gal-14, Gal-15) contain one conserved CRD domain in a single polypeptide chain, and functionally usually form non-covalently linked homodimers. The only representative of chimeric galectins is Gal-3, able to form tri- or pentamers. Another group are tandem galectins. In their structure, the two CRD domains are joined by a linker peptide (Gal-4, Gal-6, Gal-8, Gal-9, Gal-12). While β-galactose is the monosaccharide demanded for recognition, elongation of the glycoepitope increases the potency of interaction. Thus, galectins favor LacNAc disaccharide, and in particular polylactosamine repeats (Figure 5) [102,103,104]. The interesting feature of galectins is their ability to form multivalent lattices in the glycocalyx, which improves their avidity for carbohydrate ligands and contributes to stabilize cell surface receptor clusters in membrane microdomains [105]. In this form, galectins may also cooperate with DC-SIGN by cross-linking receptor glycan motifs, sometimes with mediation of other proteins (CD44) [89,106].

Galectin secretion to the extracellular space under stress condition is considered a reaction to the “danger signal”, initiating inflammatory response [107]. Increased expression and secretion of galectins is a common property of cancer cells. Many authors now consider these lectins to be important checkpoints in regulating the immune response to neoplasia. Galectin-1 is secreted by almost all types of cancer cells and is considered an important promoter of the formation of an immunosuppressive microenvironment supporting tumor development [14,99]. It was found that Gal-1 interactions with T-cell surface glycans selectively lead to the apoptosis of the effector TH1 and TH17 cells, while stimulating Tregs expansion. The same lectin also promotes the differentiation of tolerogenic dendritic cells. It has been observed (in lung cancer) that Gal-1-induced tolerant-DCs can further induce proliferation of Tregs [44].

Gal-3 is also assigned a prominent protumorigenic function. Its enhanced expression correlates with an increase in tumor metastatic activity. It has been also reported that the interaction of Gal-3 with branched core-2 O-glycan presenting extended polylactosamine units on NK cells leads to a diminished effect of their MHC-1 and impairs anti-tumor activity of these cells [44].

For a long time, galectins have been considered cancer-related, and their interactions involved in disease progress and metastasis. In recent years, research has shown that these lectins are secreted by many different cells, including those of immune system, such as macrophages and NKs. A lot of interest has been paid to the importance of galectin-mediated interactions for a healthy pregnancy [107,108,109,110,111]. In this context, galectins 1, 3, 7–10, and 13 are especially considered. They are expressed by maternal endometrium/duodenum cells, as well as the trophoblast. Their role at the maternal–fetal interface is believed to contribute to adhesion and migration of trophoblast cells, angiogenesis, and promotion of regulation against effector T-cells. Moreover, some experimental data indicate that diminished galectin expression correlates with spontaneous pregnancy loss (especially in the 1st trimester), recurrent miscarriages, and preeclampsia [107,108,109]. 

Galectins may be secreted also by vaginal epithelium, as it was described in response to *Trichomonas vaginalis* infection [112]. In this case, both suppressive Gal-1 and activating Gal-3 were detected as regulators of immune reaction. Thus, galectin secretion in the response to seminal carbohydrates may be also considered.

### 4.4. Siglecs

Siglecs belong to the superfamily known as sialic acid specific immunoglobulin-like lectins. They are transmembrane receptors expressed in numerous subpopulations of immune cells. Extracellular CRD recognizes sialic acid, a crucial monosaccharide that terminates oligosaccharides of mammal N- and O-glycoproteins (Figure 6). For this reason, siglecs are considered self-associated molecular pattern (SAMP) receptors and Siglec–Neu5Ac interaction as a physiological mechanism that prevents over-activation of the immune system [67,68,103]. Tumor cells may exploit Neu5Ac/Siglec interaction for modulation of immune cell function, supporting an immunosuppressive tumor microenvironment [113]. The human genome includes 14 genes coding the proteins belonging to the siglecs family. Regarding their evolution, siglecs can be divided into two subgroups. Group 1 includes highly conservative siglecs 1, 2, 4, and 15. Their orthologs can be found in all groups of mammals. Group 2 is referred to as CD-33-related and comprises human siglecs 3, 5–11, 14, and 16. These proteins had been subjected to rapid evolution, and therefore the repertoire of their representatives expressed in different mammals is also diversified. Group 1 siglecs are considered the inhibitory receptors. They contain an intracellular ITIM (immunoreceptor-tyrosine based inhibitors motif) or similar (ITIM-like) domain. When bound to sialoglycans, these receptors transmit an intracellular signal that blocks the synthesis of pro-inflammatory cytokines and activation of the immune system. Inhibition of Toll-like receptors (TLRs) is also involved in this signaling pathway [113,114,115,116].

The rapidly evolving Group 2 siglecs may exert both inhibitory and activating effects on the immune system. Some representatives of this group do not contain the ITIM motif. Instead, they can interact through the positively charged amino acids within the transmembrane domain with another transmembrane receptor, the DNAX protein (DAP12). This one transmits activating signals inside the cell through the intracellular ITAM (immunoreceptor tyrosine-based activation) motif. The activating siglecs are believed to have arisen as a result of evolutionary pressure exerted by pathogens and cancer cells that have “learned” to exploit hypersialylation of their glycans to suppress the host’s immune response. Moreover, the research results prove that the CD-33-related siglecs are often expressed in “pairs”—simultaneously their inhibitory and activating versions, e.g., siglec5 and siglec14, as well as siglec11 and siglec16 [65,67].

## 5. Seminal Plasma Glycome

One of the first seminal plasma proteins monitored for glycosylation pattern was acute phase lipocalin, AGP. As the protein is at least partially produced by prostate [65,117,118], AGP glycosylation may reflect pathologies within the reproductive tract. Altered sialylation and an increase in glycan branching were found to reflect the inflammation within the male reproductive tract and thus they may indirectly affect the male reproductive potential [119,120]. Glycosylation of PSA and PAP, another abundant SP glycoprotein showed prostate-cancer-related changes. Attempts to distinguish benign proliferative disease of the prostate from invasive cancer based on different glycosylation patterns were also successful [121,122,123,124].

The first comprehensive approach to evaluate the glycosylation pattern of the seminal plasma glycoprotein function, linking glycosylation changes to the biological functions, regarded glycodelin [125,126,127,128,129,130]. Glycodelin isoform S present in the seminal plasma (“male”, GdS) and produced in the female reproductive organs (isoforms GdA, C, F) differ only in their glycosylation patterns [125,131], with identical protein structure. GdS, stuck to the sperm surface and is transported to the upper parts of the female genital tract (in humans, semen components apart from gametes do not penetrate the cervical mucus), is a factor that regulates the time and site to begin the capacitation process. Thus, coating the sperm with GdS prevents their premature activation, which could ultimately lead to a reduction in the activity of male gametes before they reach the vicinity of the oocyte. Shedding of GdS from sperm surface and its replacement with the female glycoform (GdA) enables sperm activation [125,126,131,132,133,134]. Detailed glycomic analysis showed the basic differences in the glycosylation patterns: the seminal glycodelin was rich in highly mannosylated and fucosylated glycans, including Lewis^x^ and Lewis^y^ type, whereas in female glycodelin isoforms contained high amounts of sialylated complex glycans [125,126,131,134].

Another milestone in the study of the glycosylation of seminal plasma glycoproteins is related to the rapid development of mass spectrometry techniques. Seminal plasma and sperm surface glycomes have been evaluated, showing that the glycoproteins differ significantly from what was thought to be the common pattern of healthy human glycoproteins; thus, indicating unique organ specificity [135,136,137]. The most striking features were a relatively high content of high-mannose oligosaccharides, Lewis-type antigens, as well as the presence of glycans lacking a terminal sialic acid residue. These findings created a foundation for the feto-embryonic defense hypothesis, suggesting the impact of male glycome on the regulation of maternal immune response and stimulation of a tolerogenic pathway necessary for fertilization and successful development of an embryo and fetus [129,138,139,140]. 

Another study worth mentioning is that of Milutinović et al. [141], who analyzed seminal plasma prostasomes–non-cellular membrane structures that are also involved in providing the appropriate microenvironment for sperm. In their study, the authors focused on sialylation on the vesicle surface, finding it more prominent in normozoospermic than oligozoospermic men. Simultaneous ConA reactivity patterns were opposite, stronger in the oligozoospermic sample. Though the authors claim the latter feature to be mannosylation-related, in our opinion this rather relates to the glycan branching. Concanavalin A is able to bind biantennary complex glycans, while does not recognize those with three or four antennae. Marić et al. [142] in their SP N-glycome analysis identified a decreased level of biantennary bisialylated glycans (with or without fucose) accompanying increasing sperm chromatin maturity. This may be read as relatively increased glycan branching accompanying the impaired semen pattern.

Inspired by the above-mentioned findings, we decided to evaluate the frequency of mentioned changes in the context of reduced male fertility. The series of reports discussed below, exploiting MS and a set of specific plant lectins as probes, was aimed at assessing whether changes in the expression of atypical glycans may correlate with the cases of reduced male fertility. The overriding question was whether altered glycans may be ligands for female immune system lectins.

### 5.1. Does Glycome Analysis Suggest Possible Interactions with Endogenous Lectins?

The well described fact is the recruitment of immune cells to the cervical region directly post-coitum [13,143]. These cells, presenting a panel of immunomodulatory lectins as described in the former paragraphs, enter the microenvironment created by seminal plasma, which makes the direct interaction possible. The second fact that should be emphasized is the presentation of glycoepitopes, which can potentially be ligands for lectins in SP glycoproteins, and also the reports on the changes in their glycosylation pattern, observed in the patients with fertility disorders [142,144], classified as: oligozoospermia—sperm count < 16 mln/mL; asthenozoospermia—<32% of sperm with progress motion; teratozoospermia—<4% sperm with correct morphology; and normozoospermia—the semen fulfilled the WHO recommended values, while male infertility factor was suspected. So far, data on the direct interaction of seminal plasma glycoconjugates with endogenous lectins of the immune system are limited. It was experimentally confirmed that SP clusterin is a ligand for DC-SIGN [145]. With the same lectin, the reactions of mucin-type SP glycoproteins were confirmed by Milutinović et al. [146]. In the next paragraphs we discuss alterations in the glycosylation pattern in terms of their potential interaction with the immunomodulatory lectins.

### 5.2. Sialylation and Siglecs’ Ligands

As previously described, Neu5Ac is considered a “self-antigen” that can suppress the immune response when recognized with the siglecs. The mass-spectrometric analysis of SP N-glycome [147] in pooled samples of different patient groups showed differences in sialylation. Compared to the control group of men with confirmed paternity, sialoglycans were increased in normozoospermic patients with reduced fertility. Highly sialylated glycans, i.e., those presenting over two terminal Neu5Ac residues were particularly increased. In opposite, a marked reduction in sialylation and the complete absence of highly sialylated glycans were observed in the SP sample from asthenozoospermic men. Therefore, in this group a weakened siglec reactivity, and thus an increase in the effector function of the immune system, can be expected. The mentioned relationship in Neu5Ac content was confirmed in further studies of individual SP samples with sialospecific lectins of plant origin [148,149]. In the asthenozoospermic group, reactivity with *Maackia amurensis* lectin (MAA) (and therefore the presentation of α2,3 of the attached Neu5Ac) was reduced in all the analyzed glycoprotein fractions in asthenozoospermic patients. Increased presentation of sialylated glycans in the group of normozoospermic men with reduced fertility compared to the control group seems to apply to α2,6 linked Neu5Ac and only some glycoprotein bands. Interestingly, further research on PSA isolated from individual SP samples did not confirm the presence of glycans lacking terminal sialic acid [150], which were shown earlier in the complete glycome [135]. This clearly indicates that the glycosylation patterns of different glycoproteins synthesized in different glands or secretory cells of the male reproductive system may differ remarkably and thus may participate (or not) in different interactions.

### 5.3. Fucosylation, Unveiled Mannose and DC-SIGN

DC-SIGN involvement in tolerogenic adaptation of the allogenic fetus after embryo implantation and during pregnancy is also postulated. Some glycoproteins abundant in seminal plasma present highly mannosylated glycans; thus, of pathogen-associated molecular patterns (PAMPs). Many of them are also highly branched and rich in fucosylated Lewis type structures [135]. There is evidence that some seminal glycoproteins (e.g., clusterin, galectin-3-binding protein) may be direct ligands for DC-SIGN present on dendritic cells, which regulate T lymphocytes activity toward suppression the immune response for the allogenic fetus in the uterus [145,151].

DC-SIGN affinity for the two different monosaccharides, is a rare feature among the lectins. It therefore seems justified to consider that these two glycoepitopes together to allow comparison of the expression/potential presentation of ligands for the lectin. Changes in the fucosylation pattern are most commonly reported traits in the studies regarding glycosylation and disease relationship. Fucose, known as tumor-associated glycoantigen (TACA), is also regarded a “self-antigen”, and therefore its interactions may lead to the weakening of immune response. In turn, high-mannose type glycans are typical for lower organisms, including pathogens [152]. 

Xin et al. [153] in their glycoproteomic studies observed heavy fucosylation (>6 fucose units per glycan) in seminal plasma galectin-3 binding protein and clusterin identified earlier as a DC-SIGN ligand [145]. Clusterin fucosylation was also studied in detail by Janiszewska et al. [154] who aimed to distinguish fucose residues linked to the glycan with different bonds; thus, α1,6 (core), α1,3, and α1,2 (in the antennary part). This research however did not show statistically significant differences in seminal plasma of normozoospermic, teratozoospermic, astenoteratozoospermic, and oligoastenoteratozoospermic men.

In our MALDI-MS studies mentioned earlier [147], in the group of normozoospermic subfertile men an increase in the expression of mannose-presenting glycans in the N-glycome was shown, accompanied with a decrease in fucosylated glycans. This effect is not visible in the remaining infertile groups: an increase in fucosylation and a decrease in the availability of mannose has been observed. Evaluation of these effects in the context of reactivity with DC-SIGN would suggest the effect of a stronger immune system response modulated by DC-SIGN associated with idiopathic infertility, while a suppressed immune system response in men with fertility problems associated with abnormal spermatogenesis.

Again, when analyzing individual SP samples, an increase in fucosylation versus control was evident in all analyzed glycoproteins [155]. This effect was not confirmed in the studies of complete SP, in which not only N-, but also mucin-type O-glycans presenting Lewis-type antigens could be recognized by the lectin probe. The variety of fucose binding and glycoproteins that contain this monosaccharide makes it difficult to draw unbiased, constructive conclusions, especially when based on the studies that differ significantly in their methodology. Regarding mannose, glycan diversity does not seem to create such a problem. Nevertheless, the results of lectin reactivity studies are inconsistent with those obtained by mass spectrometry. In the group of normozoospermic men in many glycoproteins decreased reactivity with mannose-specific *Galanthus nivalis* lectin (GNL) was observed [156]. GNL-reactive glycans were found with a high frequency in the glycopeptide fractions resulting from PSA digestion of the semen coagulum (masses < 20 kDa) [156]. These protein fragments resulting from the natural processing of semen in the female reproductive tract seem to have a particularly diverse glycosylation pattern with rich presentation of various glycoepitopes with immunomodulatory potential.

### 5.4. T and Tn Antigens and Galactose-Specific Lectins

Some of the terminal glycoepitopes present on O-linked glycans do not differ from those presented by N-linked glycans. Thus, only truncated variants of oligosaccharides, limited to a single sugar residue (GalNAc, Tn antigen) or GalGalNAc disaccharide (T antigen) are considered unique for O-glycosylation. They are also associated with neoplastic transformation and the possible modulation of the immune response via lectin reactivity. The presence of the T antigen in the O-glycome of the seminal plasma was shown by Pang [135]. In acid-soluble glycoproteins of human seminal plasma, thus analyzed, therefore mainly mucin-like O-glycoproteins, Milutinović et al. [146] have identified the presence of T, sT antigens, Lewis^x^, sialyl-Lewis^x^, and Lewis^y^ as well as polylactosamine chains. What is important, the authors confirmed a direct interaction of SP glycospecies with the two endogenous lectins: Siglec 9 and DC-SIGN. The presence of truncated seminal plasma O-glycans was suggested in our reports [149,157] based on the reactivity with the plant VVL lectin specific for the Tn antigen. Recently, we have taken up this topic in detail, reusing plant lectins with specificity for GalNAc and Gal-GalNAc, which roughly reflect the presence of Tn and T antigens, respectively. The reactivity of seminal plasma glycoproteins, observed in Western blotting, was confirmed by isolation of lectin-reactive glycoproteins from the biological samples. Among them, fragments of semenogelin and fibronectin, lactotransferrin, prolactin induced protein (PIP), and prostate specific antigen (PSA) were identified among the isolated glycoproteins by MS methods, and in the case of the T antigen also prostatic acid phosphatase (PAP), clusterin, and Zn-α2-gp. Surprisingly, a relatively low content of MUC-6 was identified. Kovak et al., have identified direct ligands for recombinant chimeric galectin-3 in human seminal plasma, isolating them by means of Gal-3 affinity chromatography. MS analysis confirmed nine lectin bound glycoproteins; thus, angiotensin converting enzyme, aminopeptidase N, clusterin, MUC-6, PAP, PSA, ZAG, and prostaglandin H2 D isomerase [158]. The inhibitory effect of lactose and asialofetuin on Gal-3–seminal plasma glycoprotein binding indicated the interaction involving the lectin CRD domain. These findings show that human seminal plasma is also rich in potential ligands for galactose and GalNAc-specific lectins such as galectins and MGL.

## 6. Conclusions and Future Research Directions

It is worth mentioning that that the studies on male infertility have been neglected for decades and the infertility problem used to be addressed only to the female factor. Development of assisted-reproduction techniques (ART; IVF and similar) has not improved this situation, as even poor-quality semen most often gives the opportunity to acquire sperm vivid enough for the procedure. However, ART is not accessible for many infertile couples because of economical or religious reasons. What is even more important, the procedure is really aggravating for the female organism. Concluding, at the moment, our knowledge on the real reasons for male infertility is surprisingly insufficient. The current research is still directed on the search of different molecular mechanisms that may facilitate/attenuate fertilization. The methods of clinical application and possible treatment is far ahead at the moment, so it is hard to speculate on it. Of course, clinical intervention is a distant goal of the research. 

The glycoproteins in the seminal plasma present glycoepitopes that may be potential ligands for typical endogenous lectins involved in the modulation of the immune response. However, both the expression of the various lectin receptors on the surface of immune cells and the presentation of available glycospecies in the microenvironment created by the seminal plasma are extremely complex and can function in an orchestrated way that varies over time or depending on numerous external factors. The current challenge is to provide unbiased experimental data indicating for such lectin–carbohydrate interactions taking place in the human reproductive tract. Moreover, it should not be expected that such disorders will constitute a major cause of fertility disorders, rather they may be one of many mechanisms worth consideration. This fact significantly hinders research strategies: in the studies presented so far, the diversity of glycosylation patterns is clearly visible, but statistical analyses either do not confirm their significance, or remain inconsistent when studies based on different methodologies are compared. Therefore, it seems that the existing classic strategies must be modified in such a way to enable the selection among couples struggling with reproductive problems for those for whom modulation of the lectin–carbohydrate interaction axis and the resulting modulation of the immune response may actually be important. Noteworthy are—so far limited—attempts to search for actual ligands for immunomodulatory lectins. 

It is worth emphasizing that in the recent years, therapeutic interventions in the modulation of the lectin/sugar axes seems to be a promising way of developing personalized pathways that inhibit the development of neoplastic diseases. The question of whether such an approach could also apply to fertility enhancement strategies is difficult to answer today, still it raises some hope for the future. Thus, even if the efforts so far have not been crowned with satisfactory results, continued research on the glycobiology of fertilization seems justified.

## Figures and Tables

**Figure 1 ijerph-19-10489-f001:**
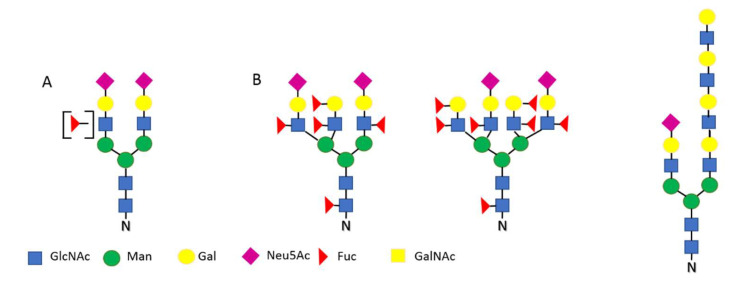
Examples of typical N-glycan structures; (**A**) dominant in healthy human serum, (**B**) common disease-related structures.

**Figure 2 ijerph-19-10489-f002:**
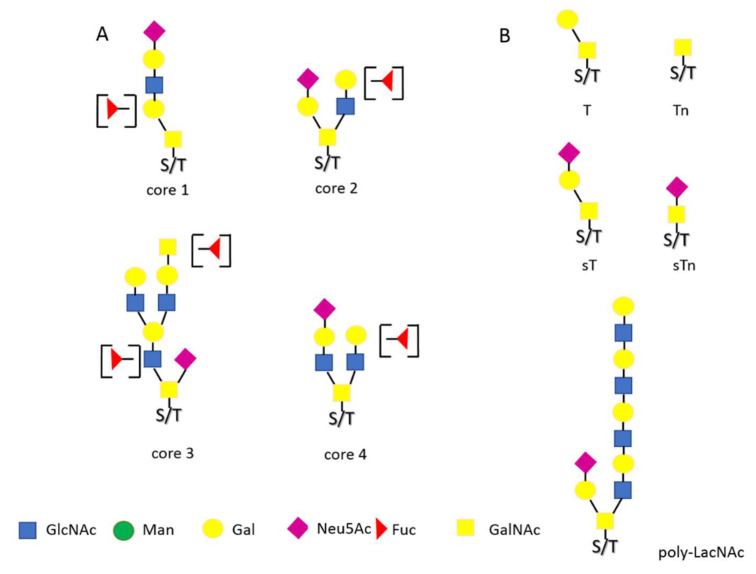
Examples of typical O-glycan structures; (**A**) dominant in healthy humans, (**B**) disease-related.

**Figure 3 ijerph-19-10489-f003:**
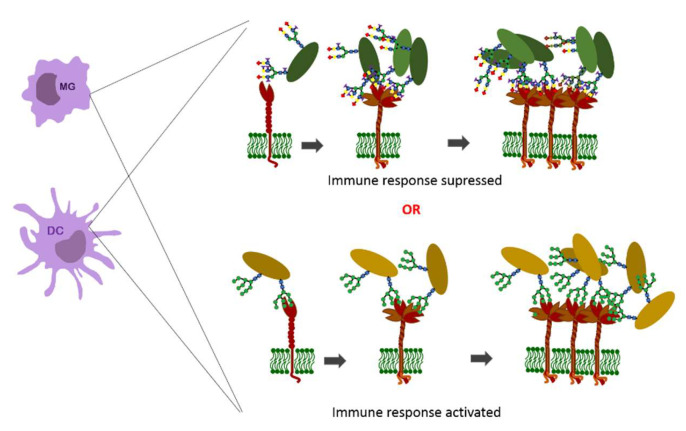
DC-SIGN presented on immune cells, their assembly in the cell membrane, and ligand binding.

**Figure 4 ijerph-19-10489-f004:**
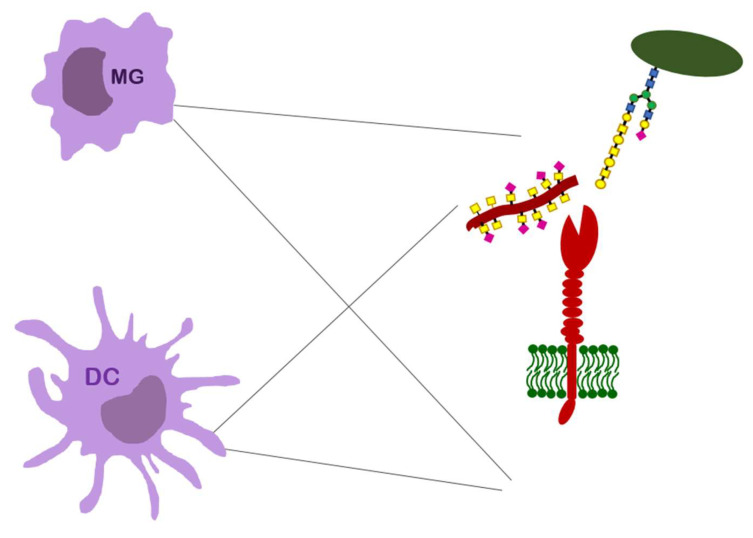
CLEC10a (MGL) presented on immune cells and its ligand binding.

**Figure 5 ijerph-19-10489-f005:**
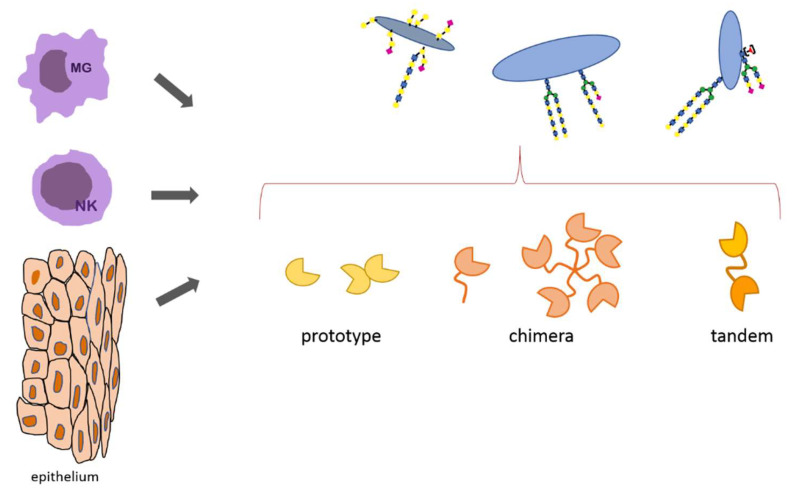
Examples of galectin secreting cells/tissues, their structures, and glycan ligands.

**Figure 6 ijerph-19-10489-f006:**
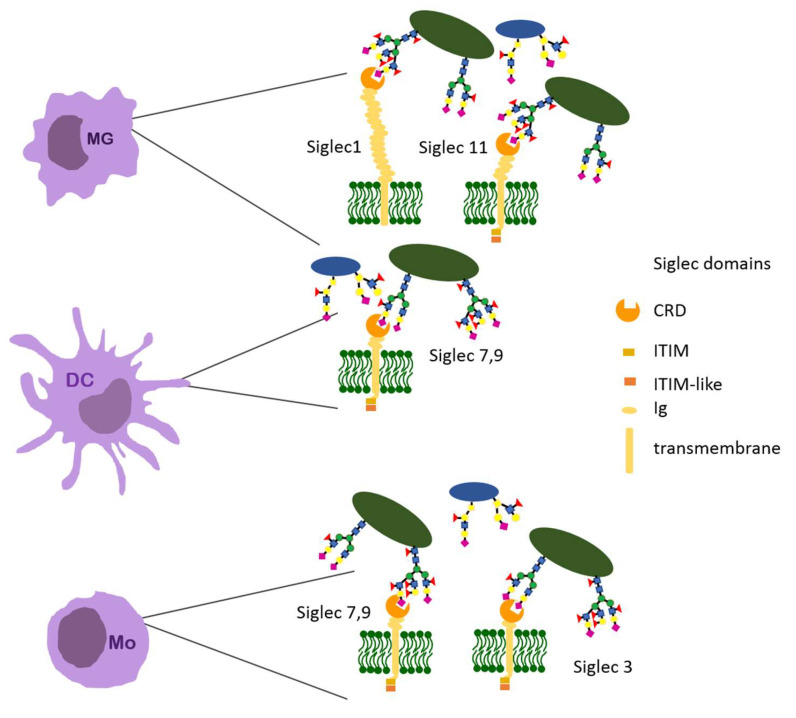
Examples of the siglecs presented on immune cells and their glycan ligands.

## Data Availability

Not applicable.

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
