# Peer review of "Seminal Plasma Glycoproteins as Potential Ligands of Lectins Engaged in Immunity Regulation"

_ijerph, 2022, doi:10.3390/ijerph191710489_

Round 1
Reviewer 1 Report
In their manuscript, Olejnik and Ferens-Sieczkowska present a review that is attempting to make the case for studying seminal plasma glycoproteins and their potential effect on the vaginal immune response to explain why some couples infertiles. At best, the observations provided are circumstantial only and the whole idea is so far not very convincing. In my opinion, the authors need to become much more critical about the current state of knowledge in the field, provide key research questions that need to be addressed in the next 5-10 years and also develop further how this would lead to potential clinical applications/treatments. Clear gaps in knowledge needs to be identified. Careful revisions by the authors to strengthen the logical thread to their argument presented in their review would hopefully lead to a clear and comprehensive review that would be relevant to the field?
The list of references looks to be current and generally complete.
Major comments
1. The ideas conveyed in this manuscript have essentially been reported already by the same corresponding author in 2021 in Biomed Res Int (https://www.ncbi.nlm.nih.gov/pmc/articles/PMC6720062/). I thought that the previous review was much more straightforward to the arguments being advanced. Yet, I see many overlaps in the written text. I would like to give the authors the opportunity here to argue as to how much this manuscript is different from the previously published review, and what new angle and/or contributions is it making now towards their research question. Also, why is not amongst the 156 cited references?
2. Throughout the text (mainly lines 60-61, 70-74, the whole sections 5. and 6.), the authors keep referring to healthy vs. diseased patients/glycans. I think for all those instances, the authors should clearly define what diseased means, and more particularly what type of disease they are referring to. This is too vague and prevents their intended message to come across clearly.
3. Section “3. Glycosylation and disease” (lines 89-108): this section is way too short. Developing it, by explaining mechanisms and providing one key example for each point listed, would really contribute to a better understanding and also help support the previous point. Also, the authors should be careful not to group all diseases together, but to treat them separately like infectious diseases, cancer, and autosomal disorders affecting glycosylation pathways.
4. Line 143: “immature DCs directs towards tolerance” I think this statement is oversimplified and misleading. First of all, an immature DC is normally sitting there, monitoring its environment through micropinocytosis, and contributes to the maintenance of homeostasis while waiting to find something. This something is usually found via the engagement of one or many PRRs, which leads to the DC to mature and become activated in order to be able to mediate their crucial immunological functions. As such, oversimplifying the role of a DC to direct towards tolerance while immature and then towards an effector T-cell response when mature (and ignoring seemingly MHC-II presentation, T-cell and B cell responses). Second, based on the references provided, this would mostly concern a role for MGL. The authors need to be critical and not over-generalize: they need to present the reported findings in their proper context as to not misguide the reader.
5. Lines 181-186: another example here of oversimplification. Again, the authors need to be much more critical, must not over-generalize and need to present the reported findings in their proper context. Here, and throughout the review, the authors present this “idea” that mannosylated glycans are pro-inflammatory and that fucosylated glycans are more anti-inflammatory. This is erroneous for many reasons. First, the mannosylated glycans are N-glycans and the fucosylated ones are O-glycans: already comparing apple with oranges. Second, the mannosylated glycans referred to are mostly fungal cell wall antigens, whereas a multitude of PRRs are being involved and are all signalling together, the sum of all which is yielding the observed response. Additionally, DC-SIGN signalling alone lead only to phagocytosis and is unable to trigger the secretion of cytokines. Instead, DC-SIGN signalling can modulate the NF-kB pathway and shift the profile of the inflammatory response produced, while other PRR-signalling (like TLRs) is necessary to activate the NFkB pathway in the first place. Third, they now compare with fucosylated Lewis-type O-glycans, which is in a completely different context of cancer, and again, DC-SIGN is not the only receptor involved in this activity. Yet, a multitude of other PRRs are involved as well, and their sum lead to a different type of response. The authors are wrong in making the claim they made here of “mannose vs. fucose”. These examples are true, but they do not relate to each other at all. Maybe they could design a project where they could experimentally compare mannosylated vs. fucosylated glycans in the same exact context.
6. Lines 291-292: Another tentative statement from the authors. This needs to be much more developed. The main question to be addressed here is: if seminal carbohydrates be sensed at the vaginal epithelium, by whom (cell types), how, and why? What is the evolutionary purpose of triggering immune responses in the vagina following coitus?
7. Section “5. Seminal Plasma Glycome”: this section constitutes the heart of the matter and for the point that the authors are attempting to make. This needs to be much more detailed, especially when it comes to what is known regarding the glycome of SP. In the context of this review, it would also be wise to compare this glycome to other bodily secretions such as colostrum, mucus, and saliva. What makes the glycans in SP unique? Here, a table or visual depiction summarizing the main glycan forms encountered in “healthy vs. diseased” SP would be much appreciated.
8. Section 5.1. and lines 370-373: it would greatly help the understanding of the reader to much further develop the immune responses of the female reproductive system, how they are conducted, or to at least provide references to great reviews on the subject. This is another key point for the argument the authors are trying to make. For example, how do those immune cells get recruited to the cervical region post-coitum? Do they reach the lumen or do they stay behind the epithelium? What type of responses are being produced and what purpose(s) do they serve? Much more details on those immune responses are warranted here, to provide the proper context on what cells and what receptors may be present here? Any data available regarding those cells types, their cytokine profile and/or the receptors they express? If unknown, a review is the perfect format to state so. All of this would help to bridge towards the potential glycoepitopes being recognized.
9. Lines 387, 390, 421 (and others if missed): please define/vulgarize the terms normozoospermic, asthenozoospermic and normozoospermic subfertile men. Not everyone is an expert in fecundation. Help the reader to better understand what those terms means, and more importantly, how those groups are defined and whether grouping them as such and producing observational studies regarding the glycome of SP can lead to the identification of targets for therapeutic interventions.
10. Section “6. Further research directions”: here, the authors should be much more critical about the overall impact of this line of research and the impact it would have compared to other known causes of infertility. Most importantly, the authors again should develop this section much further, as this is the core of their proposed theory. They should write more about how to clinically address those problems, because so far the review and the idea, as a whole, feels very hypothetic/tentative; if I was in charge of decisions regarding funding, the authors have not convinced me yet. They should also develop the therapeutic interventions mentioned at lines 487-489, as this is finally a very interesting point they are making, which deserves a full page, not a single sentence…
Minor comments
11. While the text is well understandable, this manuscript needs to be revised by a native English speaker, as there are multiple problems with sentence structure, word choice and/or prepositions.
12. In all figures, I would recommend that the use of the “a.” and “b.” text labels be made more visible by capitalizing, increasing the current font size and/or using bold.
13. Chemical nomenclature conventions that apply to glycobiology:
13.1. The “O” and “N” in N/O-glycans, N/O-glycosylation, and such, they need to be in italics. While I have found the following instances, please use the “search function” in your text processor to ensure that none is missed, including within figures when required.
Lines 20, 49, 60, 62, 78, 81, 97, 99, 107, 274, 297, 384, 431, 445, 446, 448, 450, and 453.
Line 50: It also applies to monosaccharides like N-acetylglucosamine.
13.2. Line 211: it should read as “The acetyl moiety of N-acetylgalactosamine …”
13.3. Usage of the “SA” abbreviation for sialic acid, I did find it quite confusing with other abbreviations such as SP and SAMPs. I would recommend instead that the authors use the chemical name of “Neu5Ac”, short for N-acetylneuraminic acid, which is the basic and most common form of the whole family of sialic acids. This would immediately convey to the reader that they are referring to a monosaccharide here, in the constellation of abbreviations used throughout. This needs to be done throughout the text, and do not forget the monosaccharide legends within the figures.
13.4. Line 57: Here is the first use of “sialic acid”, where the abbreviation needs to be defined for the first time.
13.5. Line 53: the saccharide unit should be written according to conventions (please refer, for example, to the textbook “Essentials of Glycobiology” by Varki et eds. (available for free on the NCBI Bookshelf)). It should be written as “Glc3Man9GlcNAc2” as on the right end the “-N” would be included to imply that it is a N-glycan. Plus the numbering needs to be formatted as subscript text.
13.6. More of a syntax rule in English, but when in a sentence you want refer to both N- and O-glycans, you need to hyphenate (“-“) the first part being used without the rest of the word. There are some instances left in the text.
13.7. Line 213: It should read as (Gal-Beta1,4-GlcNAc) where Beta corresponds to the greek symbol as it is correctly use already.
14. Line 39: Review instead of article?
15. Line 53: When referring to the “appropriate glycosylation site”, in the case of N-glycans, this motif is well known and I think the reader would benefit if the authors would mention it plainly, as it is an integral part of any basic introduction to N- and O-glycans. In the case of O-glycans, this is a much more complicated issue and as such, no general motif can be advanced for all glycoproteins.
16. Figure 1a: What about the fucose shown on the left side? Wouldn’t this fucose cap the GlcNAc and prevent elongation to LacNac or further sialylation? Also, is there a reason why the core alpha1,6-Fucose is not depicted here?
17. Line 65: Should read as “The final product of glycosylation…” instead of effect.
18. Lines 67-69: While I agree with what the authors wrote, I think that one variable missing is the protein folding and accessibility of the glycans/glycosylation sites to glycosyltransferases, as glycans play an important role in protein folding and that it also explains the macro- and micro-heterogeneity of a given glycoprotein.
19. Lines 116-117: While CLRs are named based on their calcium-dependency, it turns out that not all members of that family (such as Dectin-1) are calcium dependent. The authors should revise this sentence accordingly.
20. Line 135: the sentence regarding the immune system not working as intended, this is too vague. The authors should better define what they are trying to convey here.
21. Lines 141-142: while referring to PRRs, the authors only listed lectins and TLRs. They should also mention the NLRs and RLRs.
22. Line 148: Which one is it: Dectin-1 or Dectin-2?
23. Lines 148-149: this sentence is confusing. Which only two have been selected for the current review? Maybe list DC-SIGN and MGL as either the first or the last, so that you can refer to them that way (only the latter two were selected…”.
24. Lines 151-153: in terms of nomenclature, the field of CLRs has now adopted their own convention for reporting on CLRs. See: del Fresno C, Cueto FJ, Sancho D (2019) A Proposal for Nomenclature in Myeloid C-Type Lectin Receptors. Front Immunol 10(2098). Briefly, here DC-SIGN is also known as CD209 (no hyphen “-“) and CLEC4L. Also, it is a type-II transmembrane receptor. Finally, “superior antigen presenting cells” is not the right word. Consider using professional and also antigen-presenting needs an hyphen.
25. Line 157: this first sentence is completely unnecessary, we already know that DC-SIGN is a CLR, and that CLRs are PRRs.
26. Line 195: viral instead of virial.
27. Line 196: which is the first line of defense: neutrophils or NETs are the first line of defense? The current sentence makes no sense. But most importantly, I would rather argue that epithelial and/or mucosal physical barriers are the first line of defending against pathogens. Also, the authors should be careful, as viral glycans are none other than the host’s own glycans, that may be dysregulated as the cell producing them is infected by a virus, which took over the cellular machinery to replicate itself. I hardly see the usefulness of mentioning NETs here: either remove it, or develop it further so that it will make sense to a reader who is not familiar with the field.
28. Line 208: the title for this section should be “MGL” as this was the name used before in the text. And as with DC-SIGN (see comment 24.), MGL is also known as “CLEC10a”, not CLEX10. Also applies to line 225 (legend of Fig. 4).
29. Line 217: Because we are comparing to cancer, I would rather use the term normal instead of physiological.
30. Line 257-258: How is that cooperation between galectins and DC-SIGN even possible? This sounds like a very unlikely hypothesis that the authors came up with. Either remove it, or develop this statement much further so that it becomes solid and scientifically sound. How would that work? While they may be found at the same place and at the same time, how does a soluble galectin (now aggregated with the glycocalyx) allow interaction with membranar DC-SIGN? Or maybe DC-SIGN is reacting with the glycosylated galectins, which is unrelated to a specific interaction between those two, it’s between DC-SIGN and glycoproteins in general…
31. At lines 277-288, I would like to point to the authors that this is one the few solid, well-developed statements that they have made which would link galectins and their role in pregnancy. So far, this level of evidence if missing for CLRs.
32. Figure 6: One thing missing from this figure is the ability of Siglecs to bind cis-glycans at the surface of the same cell, that have to be displaced by the incoming glycans from another cell and/or pathogen.
33. Lines 438, 454, 459, 461, 465: many abbreviations used here are not defined anywhere in the text. Normally, to improve readability, I strongly recommend using full names, especially when mentioned only once or twice in the text. Otherwise it makes to much abbreviations to be memorized and looked-up for nothing (unlike an abbreviation used tens of times).
34. Line 465: should read as “angiotensin”.
35. Line 470: should read as “Future research directions”.
Author Response
The authors highly appreciate the detailed review of our manuscript. Below we address both the general remarks regarding the aim and scope of the review and, point by point, detailed reservations of the Reviewer.
We would also like to comment that in such a review, connecting different area of the research, there is always the issue of material selection. Considering that our potential reader is a scholar involved mainly in the reproduction and fertility area, we have decided not to provide excessive details in both glycosylation and immunity pathways. Of course this can make the reader expertized in these fields feel somehow unsatisfied. These conscious simplifications result from the scope of the IJERPH special issue to which the review is dedicated.
In their manuscript, Olejnik and Ferens-Sieczkowska present a review that is attempting to make the case for studying seminal plasma glycoproteins and their potential effect on the vaginal immune response to explain why some couples infertiles. At best, the observations provided are circumstantial only and the whole idea is so far not very convincing. In my opinion, the authors need to become much more critical about the current state of knowledge in the field, provide key research questions that need to be addressed in the next 5-10 years and also develop further how this would lead to potential clinical applications/treatments. Clear gaps in knowledge needs to be identified. Careful revisions by the authors to strengthen the logical thread to their argument presented in their review would hopefully lead to a clear and comprehensive review that would be relevant to the field?
Lectin-carbohydrate interaction attracts interest as one of the pathway involved in modulation of immune response. Numerous interesting studies address these problems in the areas if infection, cancer and autoimmunity, to mention the most important. However, this problem is rarely addressed in reproductive science, though the problem of immunity modulation is extremely important also in this context. Also, in our opinion, such studies do not reach the audience involved in reproductive science. Thus our aim was to join together some ideas in the fields of fecundity, glycobiology and immunity. Please note that within the latter area we have focused only on the selected aspects of protein-carbohydrate interactions. Though it might be read as simplification, we haven’t intended the article describing immunity mechanisms in all their complexity.
We agree that the drawback in the field is lack of experimental data that directly confirm the existence of proposed interaction. However, we intend to introduce the problem to the wider audience, especially involved in reproductive science.
I would like to mention also that the Reviewer remarks will be helpful in our future projects.
Major comments
- The ideas conveyed in this manuscript have essentially been reported already by the same corresponding author in 2021 in Biomed Res Int (https://www.ncbi.nlm.nih.gov/pmc/articles/PMC6720062/). I thought that the previous review was much more straightforward to the arguments being advanced. Yet, I see many overlaps in the written text. I would like to give the authors the opportunity here to argue as to how much this manuscript is different from the previously published review, and what new angle and/or contributions is it making now towards their research question. Also, why is not amongst the 156 cited references?
The main area of our research is glycobiology of human body fluids/secretions, and recently we have focused our interests on seminal plasma. So the former as well as the current articles regard this area and some discussed problems are similar. In the former article, however, we dealt with the issue in a wider manner, discussing the general problem of infertility and its known reasons (mostly omitted in the current manuscript), also paying attention on seminal plasma components like prostaglandins, cytokines and their role in immunity modifications. Though the aspect of glycosylation has been mentioned, it has not comprised the main focus of the former article. Also, the former one did not discussed the potential role of multitude of lectin receptors that might be involved in protein-carbohydrate interactions, which is the main issue of the current review.
The citation of the former article has been included to the reference list.
- Throughout the text (mainly lines 60-61, 70-74, the whole sections 5. and 6.), the authors keep referring to healthy vs. diseased patients/glycans. I think for all those instances, the authors should clearly define what diseased means, and more particularly what type of disease they are referring to. This is too vague and prevents their intended message to come across clearly.
We agree that the text may be confusing indeed regarding the health and disease state description. Most of the research on seminal plasma glycome refers to fertility, thus compare glycosylation pattern of fertile men (referred to as “healthy”) to the pattern observed in the semen of men with decreased fertility, most often referred to as infertile or subfertile (the use of these two terms in not strictly defined). Probably the term “of decreased fertility” is most appropriate and this one has been used in the revised text, replacing “diseased”
- Section “3. Glycosylation and disease” (lines 89-108): this section is way too short. Developing it, by explaining mechanisms and providing one key example for each point listed, would really contribute to a better understanding and also help support the previous point. Also, the authors should be careful not to group all diseases together, but to treat them separately like infectious diseases, cancer, and autosomal disorders affecting glycosylation pathways
We prefer to avoid further extending this part, as this would be duplication of some commonly known facts (also the other reviewer considered it too long). To explain our attitude: glycosylation pathway is not matrix-assisted thus results with a multitude of variants called glycoforms in both healthy organism and when we observe disease-related alterations. Thus, one cannot neither “list” the glycans present in the biological sample of healthy human (there are always several dozens of them) and contrast them to those present in the particular disease. “Disease-related” trait is the change in the relative content of particular glycoforms, so it is quantitative not qualitative. Also, it must be mentioned that even a cancer is accompanied with prolonged inflammation, so some observed changes are more prominent in advanced cancer than in basic inflammatory state. In the other words, glycosylation change can hardly be disease-specific in a qualitative meaning. That’s why we most often discuss particular glycosylation traits like “increased number of branched glycans” for example. The glycoepitopes most often affected are mentioned in section 3, page 3 and in a simplified way in the figures 1 and 2. The situation is different in hereditary diseases related to the lack of particular enzyme in the glycosylation pathway (mentioned as “autosomal” in the review), this however is beyond the scope of this review.
- Line 143: “immature DCs directs towards tolerance” I think this statement is oversimplified and misleading. First of all, an immature DC is normally sitting there, monitoring its environment through micropinocytosis, and contributes to the maintenance of homeostasis while waiting to find something. This something is usually found via the engagement of one or many PRRs, which leads to the DC to mature and become activated in order to be able to mediate their crucial immunological functions. As such, oversimplifying the role of a DC to direct towards tolerance while immature and then towards an effector T-cell response when mature (and ignoring seemingly MHC-II presentation, T-cell and B cell responses). Second, based on the references provided, this would mostly concern a role for MGL. The authors need to be critical and not over-generalize: they need to present the reported findings in their proper context as to not misguide the reader.
We agree that some statements dealing immune response are simplified. This was to some extend intended, to enable us to join-together the three research areas often regarded as not related. Thus, from the extremely complex aspects of immunity only those related to carbohydrate-protein interactions have been selected. The authors do not claim that glycosylation and its results/implications are the dominant aspects of immunity. Instead, in our opinion they have been neglected so far and this was the reason to introduce this area to the wider audience. To avoid misunderstanding the explaining remarks have been included in the text, also providing the references to the articles considering the complexity of the issue.
- Lines 181-186: another example here of oversimplification. Again, the authors need to be much more critical, must not over-generalize and need to present the reported findings in their proper context. Here, and throughout the review, the authors present this “idea” that mannosylated glycans are pro-inflammatory and that fucosylated glycans are more anti-inflammatory. This is erroneous for many reasons. First, the mannosylated glycans are N-glycans and the fucosylated ones are O-glycans: already comparing apple with oranges. Second, the mannosylated glycans referred to are mostly fungal cell wall antigens, whereas a multitude of PRRs are being involved and are all signalling together, the sum of all which is yielding the observed response. Additionally, DC-SIGN signalling alone lead only to phagocytosis and is unable to trigger the secretion of cytokines. Instead, DC-SIGN signalling can modulate the NF-kB pathway and shift the profile of the inflammatory response produced, while other PRR-signalling (like TLRs) is necessary to activate the NFkB pathway in the first place. Third, they now compare with fucosylated Lewis-type O-glycans, which is in a completely different context of cancer, and again, DC-SIGN is not the only receptor involved in this activity. Yet, a multitude of other PRRs are involved as well, and their sum lead to a different type of response. The authors are wrong in making the claim they made here of “mannose vs. fucose”. These examples are true, but they do not relate to each other at all. Maybe they could design a project where they could experimentally compare mannosylated vs. fucosylated glycans in the same exact context.
This idea probably has not been supported with appropriate references. Mannosylation indeed is a trait limited to N-glycans. In fungi this glycosylation type is dominant, however such glycans are also present in the other groups of organisms, while rare in humans. The role of such glycan in pathogen recognition is excellently reviewed by Loke et al, Emerging roles of protein mannosylation in inflammation and infection, Mol Aspects Med 51:31-55 doi: 10.1016/j.mam.2016.04.004. Regarding fucosylation, the situation is more complex, as actually it is not limited to O-glycans. Similar glycoepitopes (including Lewis type) are also present in N-glycans as well as glycolipids (gangliosides). Dense O-glycans, including fucosylated, are typical for mucins, not so abundant in seminal plasma. Fucosylated glycoepitopes have been detected in seminal plasma N-glycome (Pang et al, J Proteome Res. 2009 Nov;8(11):4906-15. doi: 10.1021/pr9001756; KaÅ‚uża et al, Carbohydr Res 2016;435:19-25, doi: 10.1016/j.carres.2016.09.009. Once again, we do not claim that DC-SIGN-sugar interaction directly triggers secretion of inflammatory cytokines, instead we indicate that lectin-sugar binding can participate in the further signaling cascade. To avoid confusion, we tried to weaken the statement in the text and replaced “mannose versus fucose” with “mannose and fucose”, as both monosaccharides may be bound by DC-SIGN.
We also agree that this is a good idea to prepare the project for experimental verification of the mentioned interaction. We work on such an idea, also basing on the material gathered and presented in this review.
- Lines 291-292: Another tentative statement from the authors. This needs to be much more developed. The main question to be addressed here is: if seminal carbohydrates be sensed at the vaginal epithelium, by whom (cell types), how, and why? What is the evolutionary purpose of triggering immune responses in the vagina following coitus?
The effect of post-coital acute inflammatory response in the cervix area, associated with the recruitment of immune cells is described in detail in D. J. Sharkey, K. P. Tremellen, M. J. Jasper, K. GemzellDanielsson, and S. A. Robertson, “Seminal fluid induces leukocyte recruitment and cytokine and chemokine mRNA expression in the human cervix after coitus,” Journal of Immunology, vol. 188, no. 5, pp. 2445–2454, 2012 and Adefuye AO, Adeola HA, Sales KJ, Katz AA. Seminal Fluid-Mediated Inflammation in Physiology and Pathology of the Female Reproductive Tract. J Immunol Res. 2016;2016:9707252. doi: 10.1155/2016/9707252. Epub 2016 Jul 3. PMID: 27446968; PMCID: PMC4947502 among others. In the latter one the authors explain „Exposure of the FMRT to SF during coitus has been shown to elicit substantial changes in the leukocyte populations within the cervix, initiating a reaction reminiscent of inflammatory response with effects that penetrate through the stratified epithelial layer and deep into the stroma of the ectocervix”
- Section “5. Seminal Plasma Glycome”: this section constitutes the heart of the matter and for the point that the authors are attempting to make. This needs to be much more detailed, especially when it comes to what is known regarding the glycome of SP. In the context of this review, it would also be wise to compare this glycome to other bodily secretions such as colostrum, mucus, and saliva. What makes the glycans in SP unique? Here, a table or visual depiction summarizing the main glycan forms encountered in “healthy vs. diseased” SP would be much appreciated.
As mentioned earlier, glycosylation pattern diversity is in most cases quantitative rather than qualitative. As described in the manuscript, the reference is usually serum glycosylation pattern. So when the relative content of particular glycoforms (overfucosylated or lacking sialic acid for example) is higher in seminal plasma of fertile men (“healthy”) comparing to the content of such glycans in the serum, such a difference indicates for tissue specificity (or secretion-specificity in this case) of the glycosylation pattern. The secretions mentioned in the review (saliva, colostrum, mucosa) are especially rich in mucins that are not so abundant in the semen, so the suggested comparison won’t be beneficial for understanding the issue.
- Section 5.1. and lines 370-373: it would greatly help the understanding of the reader to much further develop the immune responses of the female reproductive system, how they are conducted, or to at least provide references to great reviews on the subject. This is another key point for the argument the authors are trying to make. For example, how do those immune cells get recruited to the cervical region post-coitum? Do they reach the lumen or do they stay behind the epithelium? What type of responses are being produced and what purpose(s) do they serve? Much more details on those immune responses are warranted here, to provide the proper context on what cells and what receptors may be present here? Any data available regarding those cells types, their cytokine profile and/or the receptors they express? If unknown, a review is the perfect format to state so. All of this would help to bridge towards the potential glycoepitopes being recognized.
This issue is similar to p.6, so once again we can refer to the studies of Sharkey et al. and Adefuye et al. Appropriate references are provided in the manuscript. However, the very detailed information on the immune cells recruited to the cervix area is not available yet.
- Lines 387, 390, 421 (and others if missed): please define/vulgarize the terms normozoospermic, asthenozoospermic and normozoospermic subfertile men. Not everyone is an expert in fecundation. Help the reader to better understand what those terms means, and more importantly, how those groups are defined and whether grouping them as such and producing observational studies regarding the glycome of SP can lead to the identification of targets for therapeutic interventions.
Regarding the topic of the IJERPH Special Issue to which the manuscript has been submitted we actually assumed that the audience is familiar with the main abnormalities of the semen. To avoid confusion, demanded explanations have been provided in the text according to WHO directives:
oligozoospermia – sperm count < 16mln/ml
asthenozoospermia - <32% of sperm with progress motion
theratozoospermia - < 4% sperm with correct morphology
normozoospermia – the semen fulfils the WHO recommended values. Please note it does not exclude male infertility
- Section “6. Further research directions”: here, the authors should be much more critical about the overall impact of this line of research and the impact it would have compared to other known causes of infertility. Most importantly, the authors again should develop this section much further, as this is the core of their proposed theory. They should write more about how to clinically address those problems, because so far the review and the idea, as a whole, feels very hypothetic/tentative; if I was in charge of decisions regarding funding, the authors have not convinced me yet. They should also develop the therapeutic interventions mentioned at lines 487-489, as this is finally a very interesting point they are making, which deserves a full page, not a single sentence…
It is worth to mention that that the studies on male infertility have been neglected for decades and the infertility problem used to be addressed only to the female factor. Development of assisted-reproduction techniques (ART; IVF and similar) has not improved this situation, as even poor quality semen most often gives the opportunity to acquire the sperm vivid enough for the procedure. However, ART is not accessible for many infertile couples because of economical or religious reasons. What is even more important, the procedure is really aggravating for female organism. Concluding, at the moment our knowledge on the real reasons of male infertility is surprisingly insufficient. Please note that semen presenting the quality below WHO recommended values does not exclude natural conception, so it cannot be considered an unbiased reason of infertility. Thus the current research is still directed on the search of different molecular mechanisms that may facilitate/attenuate fertilization. The way to clinical application and possible treatment is far ahead at the moment so it is hard to speculate on it. Of course clinical intervention is distant goal of the research. We have completed the last section of the review with the mentioned above remarks.
Minor comments
- While the text is well understandable, this manuscript needs to be revised by a native English speaker, as there are multiple problems with sentence structure, word choice and/or prepositions.
The text has been evaluated by English language professional
- In all figures, I would recommend that the use of the “a.” and “b.” text labels be made more visible by capitalizing, increasing the current font size and/or using bold.
According to the suggestions, the capital letters have been used
- Chemical nomenclature conventions that apply to glycobiology:
13.1. The “O” and “N” in N/O-glycans, N/O-glycosylation, and such, they need to be in italics. While I have found the following instances, please use the “search function” in your text processor to ensure that none is missed, including within figures when required.
Lines 20, 49, 60, 62, 78, 81, 97, 99, 107, 274, 297, 384, 431, 445, 446, 448, 450, and 453.
Line 50: It also applies to monosaccharides like N-acetylglucosamine.
Although both versions are acceptable, mostly in glycobiology literature italics are not used. Please see Essentials of Glycobiology, fourth edition, https://www.ncbi.nlm.nih.gov/books/NBK579918/
13.2. Line 211: it should read as “The acetyl moiety of N-acetylgalactosamine …”
In this particular fragment we refer to disaccharides that may be distinguished by the lectin, thus LacNAc – GalGlcNAc, common in N-glycans, and LacdiNAC – GalNAcGlcNAc, rather rare, but appearing in both N- and O-glycans. These abbreviations are commonly used in the literature
13.3. Usage of the “SA” abbreviation for sialic acid, I did find it quite confusing with other abbreviations such as SP and SAMPs. I would recommend instead that the authors use the chemical name of “Neu5Ac”, short for N-acetylneuraminic acid, which is the basic and most common form of the whole family of sialic acids. This would immediately convey to the reader that they are referring to a monosaccharide here, in the constellation of abbreviations used throughout. This needs to be done throughout the text, and do not forget the monosaccharide legends within the figures.
To avoid misunderstanding the appropriate corrections have been made in the text and the figures.
13.4. Line 57: Here is the first use of “sialic acid”, where the abbreviation needs to be defined for the first time.
Appropriate corrections have been made and highlighted in the text.
13.5. Line 53: the saccharide unit should be written according to conventions (please refer, for example, to the textbook “Essentials of Glycobiology” by Varki et eds. (available for free on the NCBI Bookshelf)). It should be written as “Glc3Man9GlcNAc2” as on the right end the “-N” would be included to imply that it is a N-glycan. Plus the numbering needs to be formatted as subscript text.
Appropriate corrections have been made and highlighted in the text.
13.6. More of a syntax rule in English, but when in a sentence you want refer to both N- and O-glycans, you need to hyphenate (“-“) the first part being used without the rest of the word. There are some instances left in the text.
Appropriate corrections have been made and highlighted in the text.
13.7. Line 213: It should read as (Gal-Beta1,4-GlcNAc) where Beta corresponds to the greek symbol as it is correctly use already.
The text has been corrected.
- Line 39: Review instead of article?
The suggested word has been used
- Line 53: When referring to the “appropriate glycosylation site”, in the case of N-glycans, this motif is well known and I think the reader would benefit if the authors would mention it plainly, as it is an integral part of any basic introduction to N- and O-glycans. In the case of O-glycans, this is a much more complicated issue and as such, no general motif can be advanced for all glycoproteins.
Of course this remark is legitimate. Once again we intended to shorten this part of the text, still, agreeing with this reservation the change has been done
- Figure 1a: What about the fucose shown on the left side? Wouldn’t this fucose cap the GlcNAc and prevent elongation to LacNac or further sialylation? Also, is there a reason why the core alpha1,6-Fucose is not depicted here?
Fucose residue that prevents sialylation is bound to galactose unit and forms Lewis b/y antigens. This is shown in fig. 1.b. Fucose attached to GlcNAc forms Lex/a, which can be sialylated. However, similarly to core fucose, in healthy human serum fucosylation is low.
- Line 65: Should read as “The final product of glycosylation…” instead of effect.
Appropriate corrections have been made and highlighted in the text.
- Lines 67-69: While I agree with what the authors wrote, I think that one variable missing is the protein folding and accessibility of the glycans/glycosylation sites to glycosyltransferases, as glycans play an important role in protein folding and that it also explains the macro- and micro-heterogeneity of a given glycoprotein.
In this review we did not intend to provide detailed information on the glycosylation pathway. As mentioned earlier we decided to limit some information to these necessary – in our opinion – to understand the main idea of the article
19.Lines 116-117: While CLRs are named based on their calcium-dependency, it turns out that not all members of that family (such as Dectin-1) are calcium dependent. The authors should revise this sentence accordingly.
This has been changed to “mostly but not always calcium-dependent”.
- Line 135: the sentence regarding the immune system not working as intended, this is too vague. The authors should better define what they are trying to convey here.
This sentence has been change, “the immune system does not achieve the appropriate balance” was used instead of “not working as intended”
- Lines 141-142: while referring to PRRs, the authors only listed lectins and TLRs. They should also mention the NLRs and RLRs.
The suggested examples of PRRs have been included.
- Line 148: Which one is it: Dectin-1 or Dectin-2?
In this fragment we only list the examples of the members of the lectin family, so both dectins may be mentioned here. We used plural in the text.
- Lines 148-149: this sentence is confusing. Which only two have been selected for the current review? Maybe list DC-SIGN and MGL as either the first or the last, so that you can refer to them that way (only the latter two were selected…”.
The text was corrected, both lectins were listed in this sentence.
- Lines 151-153: in terms of nomenclature, the field of CLRs has now adopted their own convention for reporting on CLRs. See: del Fresno C, Cueto FJ, Sancho D (2019) A Proposal for Nomenclature in Myeloid C-Type Lectin Receptors. Front Immunol 10(2098). Briefly, here DC-SIGN is also known as CD209 (no hyphen “-“) and CLEC4L. Also, it is a type-II transmembrane receptor. Finally, “superior antigen presenting cells” is not the right word. Consider using professional and also antigen-presenting needs an hyphen.
Appropriate corrections have been made and highlighted in the text.
- Line 157: this first sentence is completely unnecessary, we already know that DC-SIGN is a CLR, and that CLRs are PRRs.
The mentioned sentence has been skipped.
- Line 195: viral instead of virial.
The spelling has been corrected
- Line 196: which is the first line of defense: neutrophils or NETs are the first line of defense? The current sentence makes no sense. But most importantly, I would rather argue that epithelial and/or mucosal physical barriers are the first line of defending against pathogens. Also, the authors should be careful, as viral glycans are none other than the host’s own glycans, that may be dysregulated as the cell producing them is infected by a virus, which took over the cellular machinery to replicate itself. I hardly see the usefulness of mentioning NETs here: either remove it, or develop it further so that it will make sense to a reader who is not familiar with the field.
According to the suggestion, this sentence has been removed
- Line 208: the title for this section should be “MGL” as this was the name used before in the text. And as with DC-SIGN (see comment 24.), MGL is also known as “CLEC10a”, not CLEX10. Also applies to line 225 (legend of Fig. 4).
The text has been corrected according to this remark
- Line 217: Because we are comparing to cancer, I would rather use the term normal instead of physiological.
Appropriate corrections have been made and highlighted in the text.
- Line 257-258: How is that cooperation between galectins and DC-SIGN even possible? This sounds like a very unlikely hypothesis that the authors came up with. Either remove it, or We put the correct references to the text.
In this form galectins may also cooperate with DC-SIGN by cross-linking receptor glycan motifs, sometimes with mediation of other proteins (CD44) [85,103].
Liu, P. et al. The Formation and Stability of DC-SIGN Microdomains Require its Extracellular Moiety. Traffic 13, 715–726 (2012).
Torreno-Pina, J. A. et al. Enhanced receptor-clathrin interactions induced by N-glycan-mediated membrane micropatterning. Proc Natl Acad Sci U S A 111, 11037–42 (2014).
- At lines 277-288, I would like to point to the authors that this is one the few solid, well-developed statements that they have made which would link galectins and their role in pregnancy. So far, this level of evidence if missing for CLRs.
Once again agreeing with this remark, we however think that presentation of these – suggested by many authors – ideas may be inspiring for further research and may lead to more solid evidence in the future.
- Figure 6: One thing missing from this figure is the ability of Siglecs to bind cis-glycans at the surface of the same cell, that have to be displaced by the incoming glycans from another cell and/or pathogen.
In our opinion too much details in the figure would make it less clear and readable. Although cis-glycan binding is important when interaction with pathogens is considered, here we decided that it may be omitted
- Lines 438, 454, 459, 461, 465: many abbreviations used here are not defined anywhere in the text. Normally, to improve readability, I strongly recommend using full names, especially when mentioned only once or twice in the text. Otherwise it makes to much abbreviations to be memorized and looked-up for nothing (unlike an abbreviation used tens of times).
We did our best to implement this recommendation throughout the text.
- Line 465: should read as “angiotensin”
The spelling has been corrected
- Line 470: should read as “Future research directions”.
This title of this section has been corrected
Reviewer 2 Report
Seminal plasma glycoproteins as Potential ligands of lectins engaged in immunity regulation by B. Olijnik et al.
This mini review is focused on the role of lectin–glycoproteins interaction in regulating fertility. This field of research is very hot and interesting for the readers.
Glycobiology, in particular the analysis of interaction among lectins and their putative ligands is opening an interesting scenario in oncology, as well as in several pathophysiological conditions
It is becoming clear that interaction of lectins with specific glycoligands is able to modulate the immunoresponse in cancer, allergic diseases, virus infection and other pathological conditions, contributing to the develop of new therapeutic strategies, which target specific lectin molecules or their ligands.
The role of glycobiology in regulating fertilization process is new and deserves to be investigated.
Although this review do not describe in detail the mechanisms regulating lectin-mediated function, it provides a general overview on specific glycosilated molecules, which could play as potential ligands for galectins, C-type lectins and siglecs (I-type lectins).
In particular the immuno-modulatory role of DC-SIGN a counter-receptor for high mannose type and higly fucosylated glycoproteins ; the role of CLEX10 (macrophage galactose-type lectin), which is specific for GalNAc residues; the immunomodulatory role of galectins-b-galactoside interaction and Siglecs – syalic acid interaction have been discussed.
Galectins, in particular Galectin-1 and 3 can be secreted by vaginal epithelium and coud play a role in fertilization process , modulating the immune-response via interaction with seminal carbohydrates.
On the other side altered sialylation and glycan branching could play a role in modulating inflammation within the male reproductive tract and thus they may indirectly affect the reproductive potential.
Investigation on the different expression of lectin molecules and the assortment of potential glycoligands in the seminal plasma represents, indeed, an interesting opportunity for scientists working on human fertility.
This contribution is well-written and has the potential to trigger new experimental work in the field of glycobiology, with potential translational value. For this reason it deserves to be published in IJERPH.
Author Response
The authors are grateful to the Referee for the time spent on the review as well as the appreciation of our manuscript
Reviewer 3 Report
The manuscript entitled "Seminal plasma glycoproteins as potential ligands of lectins engaged in immunity regulation” represents an interesting area of research and highlights the importance of continuing research about glycobiology of fertilization. References are update and makes a compilation of studies on this and the topic, which is relevant and fits perfectly with the scope of the special issue. In general terms, the manuscript is well structured and clearly written. Figures are nice and support well the information of the manuscript.
General comments:
Although I reckon that an introduction and presentation of the main features about glycosylation and glycobiology are needed, first sections of the manuscript (2-4), especially section 4, which are not directly about reproductive health, are too extensive. By contrast, section 5, which should be the most developed section, does not represent the largest part of the manuscript. Are there more studies that can be added to this section?
Specific comments:
- Title: “engaged”
- Introduction: I suggest including some sentences about the role/importance of seminal plasma in the regulation of the female immune response to paternal/fetal antigens to introduce this topic.
- Section 3: Apart from tumors, are there reproductive diseases as a result of alterations in the glycosylation patterns?
Author Response
We want to acknowledge the Reviewer’s effort to revise our manuscript. Below we address the included comments.
Although I reckon that an introduction and presentation of the main features about glycosylation and glycobiology are needed, first sections of the manuscript (2-4), especially section 4, which are not directly about reproductive health, are too extensive. By contrast, section 5, which should be the most developed section, does not represent the largest part of the manuscript. Are there more studies that can be added to this section?
In the current review our aim was to join together some ideas in the fields of fecundity, glycobiology and immunity, this latter issue limited to lectin-carbohydrates interactions. Such interactions attract interest as one of the pathway involved in modulation of immune response. However, this problem is rarely addressed in reproductive science, though the problem of immunity modulation is extremely important also in this context. We agree that the information included in section 4 does not deal directly with reproduction, still we consider it valuable to introduce the idea that numerous immune lectins may also participate in modulation of maternal immunity. As there is no information dealing directly reproduction, this is the only way to explain how the lectins work as carbohydrate receptors. Also, this is the background for the mentioned feto-embryonic defense hypothesis, assuming that interactions within the reproductive tract resemble those observed in host-pathogen recognition as well as immune evasion of cancer cells. We suppose that this is interesting to introduce these ideas to the audience involved in reproduction research. Also, we have provided more information to the section 5. These fragments are highlighted in the revised text.
Specific comments:
- Title: “engaged”
Spelling has been corrected
- Introduction: I suggest including some sentences about the role/importance of seminal plasma in the regulation of the female immune response to paternal/fetal antigens to introduce this topic.
The added fragment is as follows:
Seminal plasma components, including TGF-β and E-series prostaglandins are involved in triggering post-coital acute leucocyte reaction, thus recruitment of immune cells to the cervix area as we have briefly reviewed in our former article []. Further interactions lead to proliferation of regulatory T-cells,participating in the achievement of tolerance for male-originated antigens of sperm and embryo. According to feto-embryonic defense hypothesis [] the other mechanism to suppress maternal immunity may involve lectins present on the surface of these recruited to the cervix area immune cells and their seminal counterparts.
- Section 3: Apart from tumors, are there reproductive diseases as a result of alterations in the glycosylation patterns?
Of course cancers of the reproductive organs, both male (prostate) and female (ovaries) present the glycosylation patterns altered as a result of neoplastic transformation. Considering male reproductive health, up to our knowledge, research is focused mostly on fertility/infertility problems, referred to in this review. There is also increasing number of new and interesting information regarding glycosylation involvement in pregnancy – trophoblast implantation, recurrent miscarriages in early pregnancy, but also preeclampsia. Other interesting issues deal with endometrium receptivity and endometriosis. These problems, however, are beyond the scope of the current manuscript.
We assume that a potential reader of this article in a semen plasma journal may not be familiar with glycobiology issues, so here are some facts that may seem obvious to an expert in the field.
In fact, in this article we treat chapters 4 and 5 as equally valuable. Since data on the effects of lectins in reproductive processes are limited or absent, presenting their interactions in other processes seems to be the only way to present these important immune receptors.